# Characteristics of Intracranial Kinetic Loads When Sports-Related Concussion Occurs in Men’s Rhythmic Gymnastics

**DOI:** 10.3390/brainsci14080835

**Published:** 2024-08-20

**Authors:** Shunya Otsubo, Yutaka Shigemori, Sena Endo, Hiroshi Fukushima, Muneyuki Tachihara, Kyosuke Goto, Rino Tsurusaki, Nana Otsuka, Kentaro Masuda, Yuelin Zhang

**Affiliations:** 1Center for Education and Innovation, Sojo University, Kumamoto 860-0082, Japan; otsubo@m.sojo-u.ac.jp; 2Faculty of Sports and Health Science, Fukuoka University, Fukuoka 814-0180, Japan; n.otsuka0419@fukuoka-u.ac.jp; 3Department of Neurological Surgery, Faculty of Medicine, Fukuoka University, Fukuoka 814-0180, Japan; 4Graduate School of Sports and Health Science, Fukuoka University, Fukuoka 814-0180, Japan; ffha29@gmail.com (H.F.); mune.tachihara@gmail.com (M.T.); goto_411@yahoo.co.jp (K.G.); gd240008@cis.fukuoka-u.ac.jp (K.M.); 5Faculty of Science and Technology, Sophia University, Tokyo 102-8554, Japan; s-endo-0p3@eagle.sophia.ac.jp (S.E.); zyuelin@sophia.ac.jp (Y.Z.); 6Department of Rehabilitation, Fukuoka University Hospital, Fukuoka 814-0180, Japan; 7Faculty of Human Sciences, Kyushu Sangyo University, Fukuoka 813-0004, Japan; rino@ip.kyusan-u.ac.jp

**Keywords:** trauma brain injury, diffuse axonal injury, motion analysis, impact analysis

## Abstract

This study aimed to clarify the differences between the previously reported mechanisms of sports-related concussion (SRC) injuries without a loss of consciousness in contact and collision sports and the mechanisms of SRC injuries in our cases. Based on two videos of SRC injuries occurring during a men’s rhythmic gymnastics competition (three people were injured), the risk of SRC occurrence was estimated from various parameters using a multibody analysis and eight brain injury evaluation criteria. In the present study, the three SRC impacts that occurred in men’s rhythmic gymnastics showed significant characteristics in duration compared to previously reported cases in the contact sports. This suggests that the occurrence of SRC may have been caused by a different type of impact from that which causes SRC in contact sports (e.g., tackling). In addition, calculation of the strain indicating the rate of brain deformation suggested a risk of nerve swelling in all cases involving type 2 axonal injuries. Therefore, when reexamining sports-related head injuries, it is important to recognize the characteristics and mechanisms of SRC that occur in each different sport, as well as the symptoms and course of SRC after injury.

## 1. Introduction

The Centers for Disease Control and Prevention (CDC) reported that approximately 1.7 million Traumatic Brain Injuries (TBIs) occurred annually from 2002 to 2006 [1]. In addition, CDC has been active in efforts to prevent and respond to TBIs and concussions, including the establishment of the National Concussion Surveillance System [2].

Martland reported punch-drunk syndrome which occurs when boxers are subjected to repeated head impacts [3]. Punch-drunk syndrome was thought to affect boxers and other martial arts athletes. However, it has recently been recognized as chronic traumatic encephalopathy because a pathology similar to that of punch-drunk syndrome was identified in a retired National Football League player [4].

Concussion is defined as a “traumatically induced transient disturbance of brain function” [5] and it is classified as Diffuse Brain Injury [6]. Since 2001, the International Conference on Concussion in Sport (ICCS) has published recommendations on the risks to athletes with concussion injuries and on the prevention and response to sports-related concussion (SRC). The activities of the ICCS have thus resulted in revised rules and guidelines for SRC prevention in various sports [7,8,9].

Recent studies on SRC have also included research on SRC evaluation methods. The main assessment tool currently used in sports is the sports concussion assessment tool (SCAT). However, even if the latest SCAT6 is used, the presence or absence of SRC still cannot be reliably determined. The diagnosis and evaluation of SRC depends on the subjective aspects of the patient; therefore, various methods that can be objectively evaluated are necessary. One evaluation method used in recent years is the quantitative evaluation of the risk of SRC based on video observations [10,11,12]. It is speculated that video judgment of SRC and Internet of Things technology advances will lead to increasingly objective decision-making techniques in the future. The main kinetic parameters used to evaluate the risk of SRC are acceleration-based indicators (Severity Index [SI], Head Injury Criterion [HIC], Rotational Injury Criterion [RIC], Generalized Acceleration Model for Brain Injury Threshold [GAMBIT]) and brain deformation-based indicators (strain, strain rate, von Mises stress [vMS]). In systems for quantitatively evaluating the risk of SRC, the abovementioned indicators were utilized based on video observations. 

The occurrence of SRCs is not limited only to external forces on the face, head, and neck; they also occur when external forces on various body parts are transmitted to the head [13]. Previous reports describing the SRC analysis of videos analyzed translational external forces generated by frontal collisions such as tackles. However, while SRC occurs due to various injury mechanisms in various sports, the analysis of videos in three dimensions is important for elucidating the various mechanisms that cause SRC. The intracranial pathology at the moment of SRC occurrence has not yet been clarified, as MRI and other examinations are not practically possible. Therefore, it is extremely important to estimate the intracranial pathology at the time of injury using the above methodology.

For rhythmic gymnastics, few studies on SRC have been conducted due to the niche nature of the sport and because it is not frequently associated with SRC. However, the surveys we conducted thus far revealed that 14.3% of male rhythmic gymnasts have experienced SRC in Japan [14,15]. Men’s rhythmic gymnastics is a sport that combines the features of gymnastics and cheerleading, and the injury mechanism is suspected to be very different from SRC that occurs in contact sports and collision sports. Therefore, it is important to clarify the mechanism of SRC. 

In this study, the risk of SRC was quantitatively evaluated based on the observation of various videos of SRC injuries that occurred during men’s rhythmic gymnastics, thereby clarifying the characteristics of the intracranial environment in a three-dimensional kinetic load. This study aimed to clarify the differences between the mechanisms of SRC and intracranial injuries in our cases and those reported to date in contact sports to raise the issues identified in this study.

## 2. Materials and Methods

In this study, we included only cases of SRC without loss-of-consciousness accidents that occurred during men’s rhythmic gymnastics competitions between January 2020 and December 2022, in which the injury status was clearly confirmed by the analysis of videos. Regarding the presence or absence of SRC in the subjects, we only included patients who visited a local doctor after the accident and in which the injury was diagnosed as SRC based on imaging and examination findings.

We quantitatively evaluated the risk of SRC by carefully examining the video of each accident. A schematic representation of the evaluation method is shown in Figure 1. A motion analysis was performed using a human systemic numerical model (MADYMO ver. 7.5, TASS), and an impact analysis was performed using a human head finite element model (LS-DYNA ver.8.0, Ansys). 

In the motion analysis, models scaled based on subjects’ height and weight were employed. The initial posture of the model was determined based on the image taken immediately before the athletes collided. The initial velocity of the model was determined through trial and error by comparing the simulation with several videos obtained during the collision. The collision between athletes was analyzed using the built-in contact characteristics of MADYMO. The head posture and velocity just before the impact and the temporal history of the head acceleration during the impact were calculated for the subject. These data were then input into the human head finite element model to compute the deformation and stress within the brain caused by the impact. The human head finite element model employed for the impact analysis was developed using magnetic resonance imaging (MRI) data from an adult male. This head FE model was constructed using hexahedral solid and shell elements, with a total of 70,575 nodes and 74,420 elements. The head model consisted of the facial skin, scalp, skull, cerebrospinal fluid (CSF), cerebrum, cerebellum, corpus callosum, ventricles, and brain stem. Elastic properties were assigned to the scalp, skull, falx, and tentorium, whereas viscoelastic properties were assigned to the CSF, cerebrum, brain stem, corpus callosum, and ventricles. The validity of the model was confirmed through cadaver experiments conducted in previous studies [10].

In addition, the quantitative evaluation of the risk of SRC in 3 cases of SRC was calculated using the injury evaluation indicators of the SI, HIC, RIC, GAMBIT, maximum principle strain/strain rate, von Mises stress (vMS), and the combined indicator of maximum value of the translational acceleration of the head with vMS (a_max_ and vMS). The representative expressions of these risk predictors are shown in Equations (1)–(4); SI and HIC were calculated using translational acceleration *a*(*t*), *T* is the head impact duration, and *t*_1_ and *t*_2_ are the durations of the 2 head impacts selected to maximize HIC. The RIC was calculated using rotational acceleration *α*(*t*). GAMBIT is a predictor that uses both translational acceleration *a*(*t*) and rotational acceleration *α*(*t*).
(1)$$SI=\int_{0}^{T}{a(t)}^{2.5}dt$$
(2)$$HIC={\left\{(t_{2}-t_{1}){\left[\frac{1}{\left(t_{2}-t_{1}\right)}\int_{t_{1}}^{t_{2}}a(t)dt\right]}^{2.5}\right\}}_{max}$$
(3)$$RIC=\frac{{\left\{(t_{2}-t_{1}){\left[\frac{1}{\left(t_{2}-t_{1}\right)}\int_{t_{1}}^{t_{2}}\alpha (t)dt\right]}^{2.5}\right\}}_{max}}{10^{4}}$$
(4)$$GAMBIT={\left[{\left(\frac{a(t)}{250}\right)}^{2}+{\left(\frac{\alpha (t)}{25,000}\right)}^{2}\right]}_{max}^{0.5}$$
where *a*(*t*) and *α*(*t*) are the translational and rotational accelerations of the C.G. of the head, respectively; *T* is the time duration of the head impact; and *t*_1_ and *t*_2_ are the time periods for evaluating the maximum values of HIC or RIC. In our study, the risk of SRC was calculated using the formula proposed by Zhang et al. [12].

## 3. Results

There were two accidents involving SRC and three patients identified to have SRC during the study period. The risk of SRC was quantitatively evaluated based on a video of each accident.

Case 1: Injured patient

Athlete A: 19-year-old male (Height: 162 cm, Weight: 58 kg) 

Case 1: Injury status

During the performance of a three-stage tower combination movement with five people, the athlete at the top lost his balance and fell. At that time, he fell over the head of Athlete A on the bottom stage, thereby injuring Athlete A.

Case 1: Following the injury

Athlete A was diagnosed with SRC after undergoing diagnostic imaging upon visiting the hospital on the night of the injury because there was no improvement in his headache after the injury. With the disappearance of symptoms, he gradually returned to participate in the sport.

Case 1: Quantitative evaluation of the risk of SRC

Figure 2 illustrates the movements created based on a video of the accident. The top row shows the video, whereas the bottom row shows the corresponding actions that were simultaneously generated. Table 1 presents the posture and initial velocity of the head immediately before impact, while Figure 3 illustrates the acceleration of the head’s center of gravity. In Figure 3, the time of head impact is defined as 0 ms, and the acceleration results represent a duration of 50 ms. The translational acceleration and rotational acceleration immediately before impact were calculated from the created movements, and then the initial conditions for the impact analysis were set. 

In this case, the maximum strain value in the skull obtained by an impact analysis using the human head finite element model was 13%, the maximum strain rate value was 18.4 s^−1^, and the maximum von Mises stress value was 4.4 kPa. The mean translational acceleration in this case was 28.7 g, with a duration of 9.7 ms. The mean rotational acceleration was 0.91 krad/s^2^, with a duration of 23.1 ms.

From the above, the risk of SRC of Athlete A calculated using translational acceleration, rotational acceleration, and brain deformation was as follows: SI, 2.3%; HIC, 1.6%; RIC, 3.8%; GAMBIT, 7.8%; strain, 57.5%; strain rate, 15.3%; vMS, 62.8%; and a_max_ and vMS, 31.2%.

Case 2: Injured patient

Athlete B: 20-year-old male (height: 169 cm, weight: 62 kg)

Athlete C: 20-year-old male (height: 166 cm, weight: 62 kg) 

Case 2: Injury status

The head of an athlete (Athlete B) who was performing a rear-facing somersault contacted the head of another athlete (Athlete C) who was performing a rear-facing tuck somersault.

Case 2: Following the injury

After contact, the two patients experienced headache. Both patients exhibited no improvement in their symptoms; therefore, they visited the hospital the next day, underwent an imaging diagnosis, and were diagnosed with SRC. Their headache disappeared within 2 days (including the day of injury). With the disappearance of symptoms, they gradually returned to participate in the sport.

Case 2: Quantitative evaluation of the risk of SRC

Figure 4 illustrates the movements created based on the video of the accident, as in Case 1. Table 2 presents the posture and initial velocity of the head immediately before the impact of Athletes B and C, while Figure 5 and Figure 6 illustrate the acceleration of the head’s center of gravity. In these figures, the time of the head impact is defined as 0 ms, and the acceleration results represent a duration of 20 ms. The translational acceleration and rotational acceleration just before the impact of the heads of Athletes B and C were calculated from the created movements, and then the initial conditions for the impact analysis were set. Table 3 shows, as obtained by the impact analysis using the model, the maximum values of strain, strain rate, and von Mises stress generated in the skulls of both athletes.

In this case, the mean translational acceleration was 64.6 g for Athlete B and 49.6 g for Athlete C, while the duration was 3.3 ms for Athlete B and 3.7 ms for Athlete C. In addition, the mean rotational acceleration was 1.56 krad/s^2^ for Athlete B and 1.80 krad/s^2^ for Athlete C, and 7.6 ms for Athlete B and 7.5 ms for Athlete C.

From the above results, the risk of SRC for Athlete B in Case 2, calculated using translational acceleration, rotational acceleration, and brain deformation, was 62% for SI, 41.7% for HIC, 13.2% for RIC, 92.7% for GAMBIT, 33.5% for strain, 99.9% for strain rate, 95.6% for vMS, and 95.8% for a_max_ and vMS. The risk of SRC for Athlete C was SI: 9.7%, HIC: 6.2%, RIC: 46.2%, GAMBIT: 74.1%, strain: 46.9%, strain rate: 100.0%, vMS: 75.4%, and a_max_ and vMS: 77.3%.

## 4. Discussion

The diagnosis and evaluation of SRC is extremely important for managing the safety of athletes. Despite the latest SCAT6 reports, various methods are needed to evaluate SRCs objectively because they often depend on the subjectivity of the patient. Modern imaging approaches (e.g., advanced MR imaging biomarkers) are being used to study the intracranial effects of TBIs [16]. However, challenges remain in their clinical application. 

The present case is a contact pattern that is fully conceivable in men’s rhythmic gymnastics. However, it is difficult to analyze the impact by reproduction with either humans or objects. The evaluation method used in this study makes it possible to calculate the risk of concussion even in injury situations that are difficult to reproduce, and this method is expected to contribute to future rule revisions.

In this study, we quantitatively evaluated the risk of SRC based on a video analysis of our own cases. A similar study using the SRC risk calculation formula for contact sport athletes reconfirmed that the occurrence of SRC is strongly suspected when at least one of the SRC risk indicators exceeds 50% [12]. In our cases, the number of indicators with an SRC risk of >50% was two for Athlete A (acceleration indicators: zero, brain deformation indicators: two), five for Athlete B (acceleration indicators: two, brain deformation indicators: three), and four for Athlete C (acceleration indicators: one, brain deformation indicators: three).

In the present and previous study [10,11], when looking at the indicators with an SRC risk of >50%, Case 2 exhibited a tendency for indicators of >50%, similar to the cases of SRC caused by tackling in American football. However, for Case 1, none of the characteristics were similar to those observed in the cases of injuries in American football and judo (Table 4).

I. Differences in the acceleration in the SRC risk calculation formula between our cases

In Case 1, there was no indicator based on an acceleration that exceeded 50%, and the calculated risk was lower than that of Athletes B and C in Case 2 (Table 4). The formula used to calculate the SRC risk in this study may influence the fact that the risk of SRC was calculated based on the indicator related to acceleration. The calculation formula used in this study was based on American football, in which the SRCs occur due to translational external forces such as tackling. Therefore, cases with low acceleration might not be considered in this calculation formula, and it is presumed that the risk of SRC was calculated in our cases (Case 1), which involved a low-acceleration impact. In the future, we believe that a more accurate SRC risk calculation formula can be established by collecting a large number of cases in which SRC occurs in low-acceleration impacts, such as that observed in Case 1, and thus reflecting the results obtained from the analysis of these cases in the SRC risk calculation formula.

II. Considerations concerning mean translational acceleration and duration

The mean translational acceleration of Case 1 was 28.7 g, with a duration of 9.7 ms. In Case 2, Athlete B was 64.6 g and Athlete C was 49.6 g, while the duration was 3.3 ms for Athlete B and 3.7 ms for Athlete C.

Comparing the cases of these three patients to previous studies [17,18], Case 1 exhibited a lower impact intensity and a shorter duration of impact relative to other sports. As a result, the impacts showed no similarity to those seen in other sports. For the two patients in Case 2, the impact was not similar in intensity to the cases of American football (98 ± 28 g) [17], and when compared to judo, the impact values of the two athletes were greater than those experienced in osoto-gari (41.0 ± 2.6 g) [18]. Additionally, it was revealed that the duration of Case 2 was shorter than that previously reported in judo (approximately 25 ms) and American football (15 ms) [17,18]. Therefore, as in Case 2, the impacts showed no similarity to those seen in other sports. These findings suggest that in men’s rhythmic gymnastics, SRC may occur because of impacts that are clearly different from those in contact sports in terms of the relationship between translational acceleration and duration.

III. Considerations concerning the mean rotational acceleration and duration

The mean rotational acceleration of Case 1 was 0.91 krad/s^2^, with a duration of 23.1 ms. Furthermore, in Case 2, the mean rotational acceleration of Athlete B was 1.56 krad/s^2^ and that of Athlete C was 1.80 krad/s^2^, while the duration was 7.6 ms for Athlete B and 7.5 ms for Athlete C.

In comparison to previous studies, the rotational accelerations of the three athletes were smaller than those obtained in American football (6.4 ± 1.8 krad/s^2^) [17]. These were also lower than the values experienced in judo in osoto-gari (3.3 ± 0.2 krad/s^2^) [18]. Additionally, when focusing on duration, Case 1 had a shorter duration than that observed in judo, and Case 2 had the shortest duration compared to American football (15 ms) and judo (osoto-gari: approx. 30 ms) [17,18]. These points suggest that in terms of impact, the relationship between rotational acceleration and duration, patients in Case 1 and the two patients in Case 2 may have suffered SRC due to impacts that differ from those in contact or combat sports.

IV. Considerations concerning impact duration

There have been many reports on SRC, but most involved SRC occurring in contact and collision sports. In reality, the mechanisms of SRC and the circumstances of injury vary from sport to sport. Although several cases of SRC during men’s rhythmic gymnastics competitions were investigated in this report, it is assumed that there are at least two major mechanisms of SRC associated with this sport. One involves mistakes during group gymnastics, as in Case 1. The other involves contact during rotation-type techniques, such as a backflip or flip, as in Case 2. In Case 2, the durations of both translational acceleration and rotational acceleration were shorter in comparison to the other sports. However, the duration of rotational acceleration in Case 1 was longer than that in American football. The reason for the longer duration of impact in Case 1 was due to the fact that the injury was caused by an upward fall. Therefore, the situation was one in which the falling person impacted the injured person’s body and the duration of contact was considered to be long. In general, in a group routine, such as in Case 1, it is suspected that SRC occurs in athletes who fall from a high position; however, in this case, SRC occurred in an athlete who was at the base of the group. Therefore, it is necessary to take measures so as to not overlook SRC in sports such as men’s rhythmic gymnastics and cheerleading where routines like the one in Case 1 are performed. The competition should be immediately suspended and all athletes involved in the accident should be assessed for SRC when a comparable incident occurs.

V. Considerations concerning strain

Impacts to the brain include coup injury, contrecoup injury, and shearing injury. The type of impact depends on the circumstances of the sports injury. SRC causes various brain dysfunctions, even though there are no obvious intracranial lesions. Diffuse brain injuries, including SRCs, are mainly caused by external forces that may deform the brain. Severe brain deformation can cause diffuse axonal injury. Thus, previous studies have also examined the rate of deformation that occurs when SRC occurs. When the brain is subjected to rotational or shaking forces due to TBI, the white matter, the part of the brain that contains many axons, is extensively distorted and damaged. This is called diffuse axonal injury. According to Maxwell et al., when a brain distortion of 10–15% occurs, nerve swelling occurs in type 2 axonal injury [19]. Incidentally, type 2 axonal injuries are classified into three categories: 5–10% for stage 1, 10–15% for stage 2, and 15–20% for stage 3 [19]. In Case 1, type 2 axonal injuries accounted for 13% of the cases. In Case 2, the brain distortion of Athlete B was 11%, while that of Athlete C was 12%. All three patients fell under stage 2 in this study. Since all three cases in which SRC occurred had the same degree of distortion, it can be inferred that SRC occurring in men’s rhythmic gymnastics was caused by a strong impact, even when no loss of consciousness occurred. 

From the results of this study, it can be inferred that when athletes contact each other in men’s rhythmic gymnastics during rotation-type techniques, the impact is stronger than expected, thus causing a large strain on the intracranial area, which is likely to result in SRC. Therefore, when a similar accident occurs, it is important to stop competition immediately so that athletes suspected of having SRC can receive immediate treatment at a medical Institution.

VI. Limitation

The three cases in this study were considered to have potentially different impacts from other sports. However, due to the small sample size for comparison purposes, statistical significance could not be determined. Therefore, a larger number of cases should be collected in order to conduct further comparative studies in the future.

## 5. Conclusions

For the three patients evaluated in this study, the duration of SRC in men’s rhythmic gymnastics distinguished it from SRC cases in other contact sports, thus suggesting that the SRC in gymnastics may have been caused by a distinct type of impact from that experienced during tackling and other moves in contact sports. In addition, even in the absence of a loss of consciousness in SRC, the calculation of the strain indicating the deformation rate of the brain suggested the risk of nerve swelling in type 2 axonal injuries in all cases. In summary, concussions in such sports as men’s rhythmic gymnastics may have a low risk of occurrence, but they may be more likely to be severe.

Moving forward, it is crucial to examine additional cases of similar SRC injuries to enhance the precision of risk calculations and create a safe sporting environment that makes it possible to avoid severe accidents.

## Figures and Tables

**Figure 1 brainsci-14-00835-f001:**
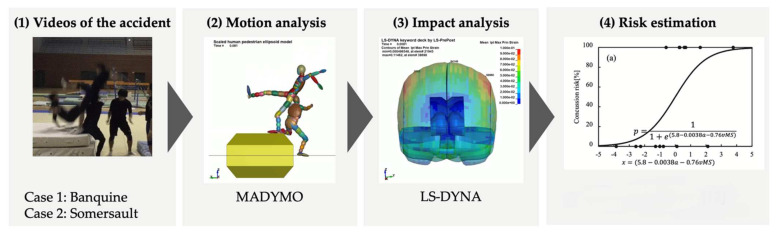
Concussion risk evaluation system, Zhang Y. et al. 2018 [12].

**Figure 2 brainsci-14-00835-f002:**
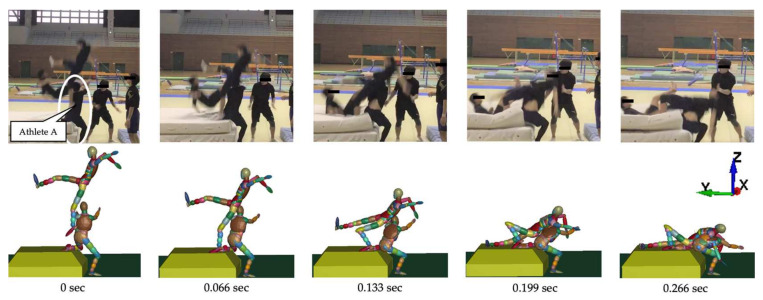
The falling motion of players reconstructed based on the accident video (Case 1).

**Figure 3 brainsci-14-00835-f003:**
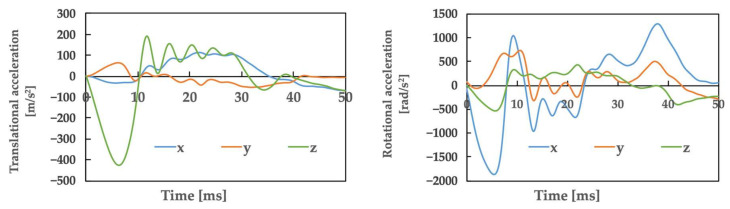
Athlete A: The accelerations obtained from motion analysis.

**Figure 4 brainsci-14-00835-f004:**
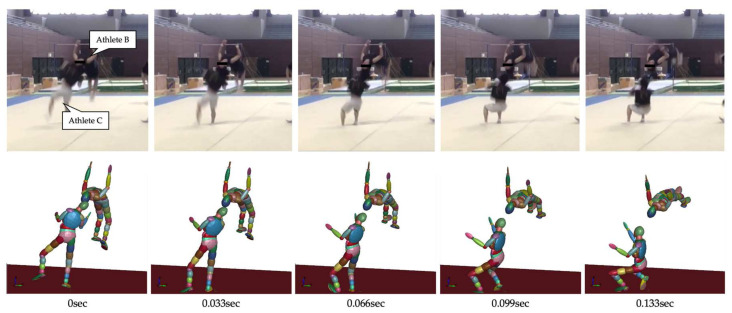
The falling motion of players reconstructed based on the accident video (Case 2).

**Figure 5 brainsci-14-00835-f005:**
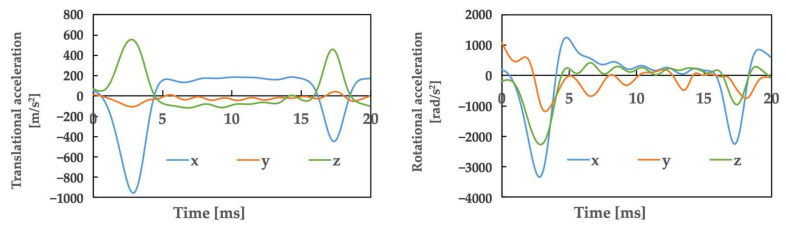
Athlete B: The accelerations obtained from motion analysis.

**Figure 6 brainsci-14-00835-f006:**
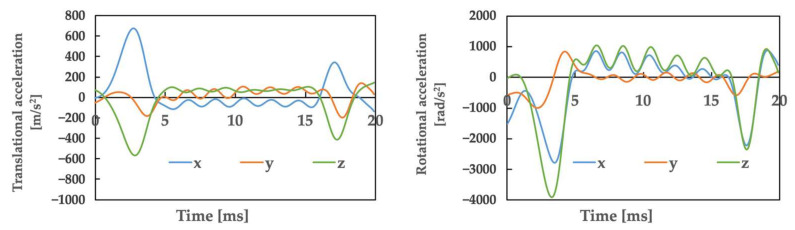
Athlete C: The accelerations obtained from motion analysis.

**Table 1 brainsci-14-00835-t001:** The posture and initial velocity of the head just before impact.

	x	y	z	Resultant Value
posture [rad]	−0.037	0.152	−0.038	-
translational velocities [m/s]	0.21	−0.63	−1.16	1.34
rotational velocities [rad/s]	0.79	0.97	−2.23	2.56

**Table 2 brainsci-14-00835-t002:** The posture and initial velocities of Athlete B and C’s head just before impact.

	Athlete B				Athlete C			
	x	y	z	Resultant Value	x	y	z	Resultant Value
posture [rad]	2.820	0.734	0.165	-	2.210	−2.842	0.193	-
translational velocities [m/s]	5.00	−0.01	−1.65	5.27	1.49	−2.64	−2.72	4.07
rotational velocities [rad/s]	−0.11	5.63	0.19	5.64	2.93	8.36	9.51	13.0

**Table 3 brainsci-14-00835-t003:** Maximum value of evaluation indicators related to brain deformation of accident Case 2.

	Strain (%)	Strain Rate (sec^−1^)	vMS (kPa)
Athlete B	11	56.3	6.2
Athlete C	12	120.8	4.84

**Table 4 brainsci-14-00835-t004:** Evaluated injury risk of SRC based on kinetic parameters.

	SI	HIC	RIC	GAMBIT	Strain	Strain Rate	VMS	Amax/vMS
Athlete A	2.3	1.6	3.8	7.8	57.5	15.3	62.8	31.2
Athlete B	62.0	41.7	13.2	92.7	33.5	99.9	95.6	95.8
Athlete C	9.7	6.2	46.2	74.1	46.9	100	75.4	77.3
American football: Case 1 (Aomura et al., 2016) [10]	98.6	98.8	4.21	97.6	37.3	84.5	61.5	
American football: Case 2 (Aomura et al., 2016) [10]	99.9	99.9	1.41	99.5	25.5	60.2	55.5	
Judo (Zhang et al., 2017) [11]	66.6	72.8	1.1	39.4	27.3	29.5	60.0	

## Data Availability

The original contributions presented in the study are included in the article; further inquiries can be directed to the first author or corresponding author.

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
