# Peer review of "Characteristics of Intracranial Kinetic Loads When Sports-Related Concussion Occurs in Men’s Rhythmic Gymnastics"

_brainsci, 2024, doi:10.3390/brainsci14080835_

Round 1
Reviewer 1 Report
Comments and Suggestions for Authors The manuscript presented from Shunya Otsubo et al., entitled "Characteristics of intracranial kinetic loads when sports related concussion occurs in men's rhythmic gymnastics", is interesting and original. However there are different critical point to be addressed: - The introduction must be improved, the authors must include more reference. - The authors must motivate why their experimental approach is innovative and original, it is not clear. - The authors must improve the statistical analysis, it is not clear how they counted. - The authors must improve the discussion also comparing with previous studies.Author Response
Please see the attachment.

Reviewer 2 Report
Comments and Suggestions for Authors
The manuscript is well-written and presents a comprehensive study on a significant topic, highlighting the scientific importance of the findings in advancing our understanding of the subject matter, which is kinetic load that could impact further research in TBI. However, there are a few minor errors that need correction to improve clarity and coherence. Typographical errors such as "Qutient" to "Quotient" and "assesment" to "assessment" should be corrected, and consistency in terminology and formatting should be ensured throughout the manuscript, particularly in figure legends and table headings. Clarify ambiguous sentences to improve readability, and review the abstract and conclusion to succinctly summarize the key findings and implications of the study without unnecessary repetition. Verify that all references are correctly cited and formatted according to the journal's guidelines, and ensure there are no missing or incorrect references. Additionally, review the manuscript for grammatical errors and awkward phrasing, paying attention to subject-verb agreement, punctuation, and the use of articles. These minor revisions will enhance the clarity, readability, and overall professionalism of the manuscript, ensuring it meets the high standards of academic publication.
Comments on the Quality of English LanguageSee above
Reviewer 3 Report
Comments and Suggestions for Authors
Dear Authors,
Thank you for conducting this study.
Although the title is novel and interesting, the lack of cases makes the conclusion illogical. The study is well-written. Just please add limitations at the bottom of the discussion. Sorry if my comment is disappointing.
Sincerely
Round 2
Reviewer 1 Report
Comments and Suggestions for Authors
The authors improved the quality of the manuscript that now is a case report
Reviewer 3 Report
Comments and Suggestions for Authors
Thank you for making revisions.
Unfortunately, I still think the result is invalid due to a lack of cases.
Sincerely